# Helping People Predict the Outcome of Robotic Pouring Behaviors with Augmented Reality

Andre Cleaver
andre.cleaver@tufts.edu
Tufts University
Medford, Massachusetts, USA

Reuben M. Aronson
reuben.aronson@tufts.edu
Tufts University
Medford, Massachusetts, USA

Jivko Sinapov
jivko.sinapov@tufts.edu
Tufts University
Medford, Massachusetts, USA

## ABSTRACT

People effortlessly manipulate fluids due to their intuitive understanding of fluid dynamics, while robots struggle with complex fluid dynamic calculations, particularly in tasks like pouring. To enhance assistive robots in such tasks, we propose involving users in correcting and providing feedback by visualizing the planned pouring trajectories before they are executed. This paper investigates whether people can predict robotic pouring outcomes and make adjustments to minimize spills, using visualization devices like augmented reality. In a human-participant study, participants evaluated and adjusted robot pouring behaviors of unique configurations for various source containers. Results highlight the effectiveness of visualization tools such as augmented reality headsets, as well as traditional 2D display, especially with specific pouring parameters, and users noted their benefits in open-ended responses. This research illuminates the potential for human-robot collaboration in fluid manipulation tasks, with visualization tools reducing spills in robot-controlled pours.

## KEYWORDS

Augmented-Reality, Robotics, Fluid-Manipulation, Pouring

**ACM Reference Format:**
Andre Cleaver, Reuben M. Aronson, and Jivko Sinapov. 2024. Helping People Predict the Outcome of Robotic Pouring Behaviors with Augmented Reality. In *Proceedings of Companion of the 2024 ACM/IEEE International Conference on Human-Robot Interaction (HRI '24 Companion).* ACM, Stockholm, SE, 10 pages. https://doi.org/XXXXXXX.XXXXXXX

## 1 INTRODUCTION

Individuals accumulate years of experience observing both their own and others' actions in manipulating fluids. This extensive experience equips us with an internal physics engine, allowing us to anticipate the results of a pour and intuitively identify necessary adjustments during the pouring process to prevent spills [5, 31]. However, it remains uncertain whether individuals can accurately predict the outcomes when a robot is manipulating fluids, particularly when they lack a precise mental model of the robot's

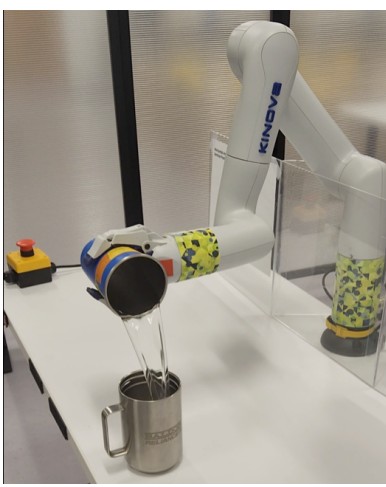

**Figure 1: The robot used in our study, shown pouring water from a source to target container. In our experiments, participants were first tasked with predicting the outcome of a pouring action (spill vs. no spill). Participants were also tasked to adjust the parameters of the pouring action to minimize spills.**

capabilities upon initial encounter. Users employing an assistive robotic arm for pouring a drink may find themselves at risk of spills due to unfamiliarity with the robot's behavior.

Pouring fluids from container to container is a challenging task in robotics and is still considered an open problem requiring advanced simulators for manipulation learning [12]. Because the fluid dynamic calculations are often too difficult to solve, robots may still not apply the appropriate actions to avoid any spillage. We can leverage human ability to predict pouring outcomes and enable people to improve a robot's pouring capabilities by modifying parameters related to the manipulation process to minimize the chances of a spill. For example, a user may want to adjust the rotational speed if they foresee that the planned motion may result in a spill.

In this paper, we first want to determine to what extent visualization tools enable users to predict the outcome of a pouring action, and second, determine whether such tools enable users to adjust the planned behavior as to prevent a spill. Augmented Reality (AR) has been used to visualize a digital twin of a robotic arm manipulator simulating the pouring action corresponding to parameter changes made by the user.

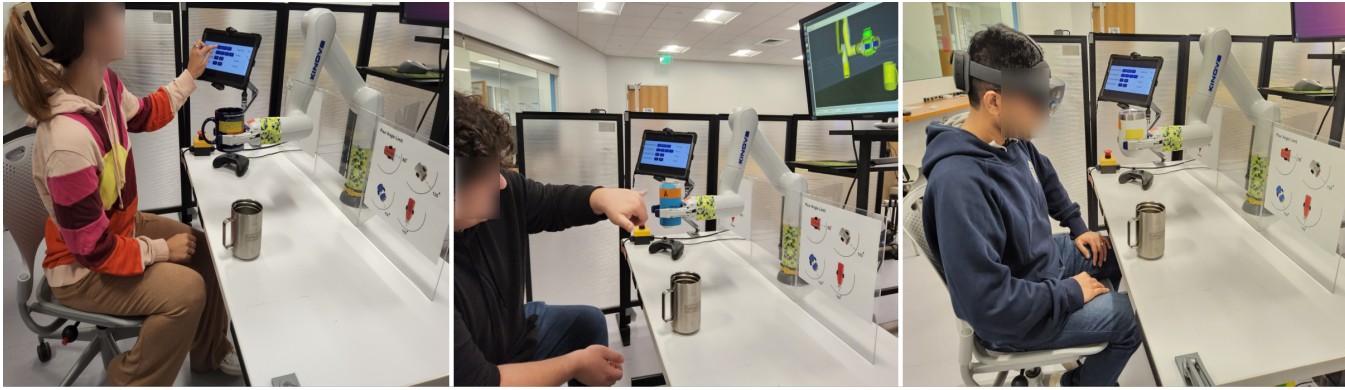

**Figure 2: Image shows the three conditions in our study. (Left) A participant in the No Visual condition (CT), shown interacting with the UI panel to change the pouring parameters, (Middle) A participant in the 2D Display condition (RZ) with a display monitor positioned above the arm, (Right) A participant in the augmented reality (AR) condition wearing a Hololens2.**

The users with an assistive robotic manipulator in this case can leverage AR to provide more contextual information to help them minimize the chances of a spill before the robot executes the pour action. Another justification for utilizing an AR device lies in the fact that the conventional approach to visualizing data necessitates users to possess a display screen. This ensures the presentation of a current representation of the real-world environment, encompassing the position and orientation of the digital twin of the target container, a feature distinct from augmented reality. We hypothesized that people can better predict the outcome of the pouring action by visualizing the digital robot simulating different pouring parameters and anticipate outcomes.

We investigated how visualizing the planned trajectory of parameter changes, such as the robot's wrist performing a pouring action at different speeds, can empower people to adjust parameters of robotic pouring actions to reduce spills. We specifically ask, "Can people make reasonable predictions for a robot pouring liquids better by observing a visualization of the planned action in AR and 2D displays before it is actually executed?", "Does superimposing a digital twin of the robotic manipulator over the real robot simulating pouring actions in augmented reality enable people to make more accurate guesses for the outcome of a robot pouring liquids?", and "Do users prefer AR over 2D displays when controlling the pouring parameters of a robot before executing a pour?". To answer these questions, we conducted a human-participant study where participants were instructed to assist a robot transferring fluids by manipulating the robot's pouring parameters. We hypothesized:

> **H1**: Visualizing the actions with digital twins enables users to better predict the likelihood of a spill caused by a robot manipulator pouring liquids from a source container. (control vs. AR/ 2D Displays)
>
> **H2**: Users with augmented reality to visualize robotic parameter changes will result in selecting parameters **more accurately (H2a** ) leading to **less spillage (H2b)**. (AR vs. control/2D Displays) The visual feedback provided by augmented reality will enable users to make more informed decisions with **less uncertainty (H2c)**, resulting in improved

control over the pouring process and minimizing the occurrence of spills.

> **H3**: Users will have more successful pours involving wider-rimmed containers compared to narrow-rimmed containers for all three visual conditions.

We gathered a combination of objective and subjective measures that showed spill prediction remains a challenge even with the help of visualization tools; however, AR was useful in reducing spills when setting pouring configurations for various containers especially when setting the pour angle limit and horizontal position although participants remained unsure of their decisions. Overall, participants seemed to objectively perform better with narrow containers although the difficultly in the type of container is a matter of speculation. Our AR visualization setup could further extend to other applications that require robotic systems to manipulate sensitive substances that are deemed too dangerous for a human to handle on their own.

## 2 RELATED WORK

Prior works related to robotic pouring have focused on developing algorithms [16, 29, 30, 37], fluid dynamics calculations [22], teaching from demonstration [25, 44, 52], and reinforcement learning [48] approaches to achieve successful pours. Although these recent works have shown promising solutions, these robotic systems require expertise to operate autonomously. Non-expert users relying on a robotic assistive device may not feel comfortable having the robot manipulating fluids unless they can supervise the process and intervene in the event the robot miscalculates a pour. In our work, we want to develop a visualization tool to enable users to make adjustments to a robots pouring behavior to enhance their confidence towards the robot making more successful pours rather than risk the robot miscalculating a pour, resulting in a spill.

Research groups have also worked to build the robot's own internal model of intuitive physics through observations of a dynamic environment [1, 35, 51] similar to how people developed this capability at an early age [23]. Studies have utilized game engines to gain insights into the core intuitive physics [49]. In our study, we

explored how users can leverage their own internal visual intuitive physics engine to determine the ideal location and rotation speed to make a robot perform a pour action with the least amount of spillage.

Explainable robotics is an emerging field that aims to enable robotic systems to effectively communicate their actions, decisions, and perceptions of their environment to people which otherwise becomes a "black box" [45]. AR technology renders graphical information over the real-world with various camera devices [4, 20] and has been widely used in the Human-Robot Interaction community [2, 13, 32, 47] in applications developed for educational [7, 15], training [17], and maintenance/debugging purposes [26, 27, 34]. Specifically, research groups enabled robots to convey motion intent [14, 19, 21, 41, 43, 50], cognitive and sensory data [8–10, 36], safety information [18, 33], and affordances [39]. Relevant works have integrated robotic systems with digital twin (DT) technologies [53] to enable users within the manufacturing settings to monitor, plan, and control robots [6, 42, 43]. Work by Arboleda *et al.* focused on improving task performance of human-robot collaborative pick-and-place tasks by improving a participant's distance perception with AR [3]. Here, participants controlled a robotic arm manipulator to grasp solid objects by pressing buttons that moved the robot's end-effector and gripper orientation. Similar to their study, our participants controlled a robot to reposition the robot's end-effector to an ideal location near the target container prior to a pour action. In our study, we have participants set the robot's pour speed which cannot physically play out because such feedback will force a pour. We want to determine if participants make use of the AR visual feedback when they adjust the robot's pouring parameters to result in a minimal spill. To our knowledge, there is a lack of work exploring a human's internal physics engine with AR, as well as leveraging AR to predict robotic pouring outcomes.

One notable study by Hashimoto *et al.* [24] introduced an augmented reality device for robot control. In this study, participants interacted with a touchscreen interface that enables them to control a mobile-manipulator to complete a pick-and-place task. Using their developed augmented reality device, *TouchMe*, participants individually controlled parts of the robot to accomplish the task of placing a bottle into a trash bin. The task of transporting a bottle into a trash bin is relatively simple compared to the complexity of pouring a fluid to a target container, and their task remained the same without any changes to the environment. The participants were also evaluated only on the three scheduling methods for controlling the robot. In our study, we tasked participants to first observe the robot to determine the outcome of the robot's pour, then manipulate the robot's pouring behavior to minimize any chance of a spill.

Another notable work by Stilman *et al.*[46] developed an augmented reality tool to help resolve any ambiguities that could occur with any complex robotic system. Because a robot could contain multiple subsystems that interact within one another, it can more difficult to determine what part of the robot's visual, navigational, or controller pipeline that is problematic in the event of an undesired outcome. By visualizing the robot's perceived ground truth, the researchers hope to identify any experimental failures and address them prior to any robot actions. The technical demonstration does not show that a user study was conducted to evaluate the

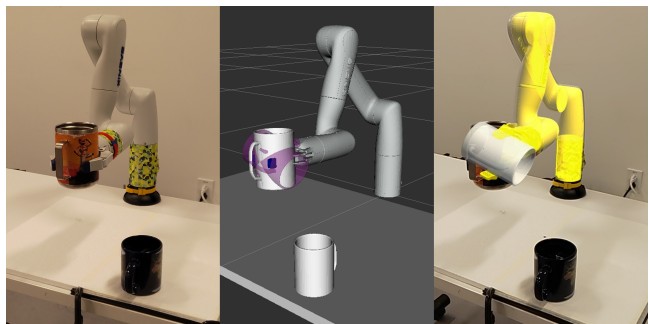

**Figure 3: Visualization of the three conditions in our study: (Left) No Visuals/Control(CT), (Middle) RViz/2D Display(RZ) with a model of the environment, (Right) AR from the prospective of the participant. Screen capture from a Hololens2. The robot and source container digital twins are superimposed over the physical robot and container respectively.**

effectiveness of their system. In our paper, we intend conduct a human-participant study to evaluate how visualization tools can help users identify the problem in a robot's planned actions and and essentially reprogram the robot's action.

## 3 METHODOLOGY

We conducted a 3 x 10 mixed-design study. Participants were randomly assigned to two out of three visualization conditions where each participant switched to their second assigned condition halfway through the experiment. All participants underwent 10 rounds of different common source containers divided into 2 groups where the order of conditions were counterbalanced. Participants were instructed to help a robot pour liquid from one container to another by adjusting the robot's *pouring* parameters. Our independent variable was the method of visualizing the robot's pouring behavior.

### 3.1 Procedure

In 10 rounds, all participants completed a two-part task to assist a 6 degrees of freedom robotic manipulator with pouring water from various source containers to a target container without causing a spill. The distance between the source and target container along with the rotational angular limit and rotational velocity all determine whether or not a spill will occur. Prior to the study, the research coordinator curated a set of 10 pouring configurations that result in either a confirmed spill or a successful pour. Here, a spill occurs when water falls outside of the target container. The process of determining these configurations involved a rigorous series of trial-and-error iterations. Initially, for each container, we meticulously explored a configuration space until identifying one that consistently led to a successful pour without spillage. Subsequently, we systematically varied parameters, documenting outcomes to discern their influence on spillage occurrence. To ensure robustness, we meticulously replicated these configurations, verifying their consistency in both success and failure scenarios. Participants were notified that all containers possessed a pour configuration resulting in no spills. Moreover, certain container's pouring configurations

were intentionally manipulated to induce spillage, prompting participants to adjust specific parameters for spill avoidance.

The pouring action by the robot involved only a single joint rotation of the robot's wrist with the source container grasped by the robotic gripper. The pour action is constrained by the angular limit and rotational velocity. The angular limit is how far the robot will rotate its wrist, and the rotational velocity is how fast the robot will rotate its wrist. The position of the robotic gripper is also constrained along the x-z plane and could only move either horizontally or vertically. Translation along either axis occurs in discrete 1.5 cm lengths. For each round, the research coordinator placed a source container with water in the robot's gripper at marked heights and set the pouring parameters for the robot before the pour action. The target container remained in the same location throughout the study. The amount of water in each container was approximately half the container.

The first part of the task involved prediction of the robotic pour outcome given the source container and assigned pouring parameters, and the second part involved adjusting the robotic pouring parameters to achieve a pour without causing a spill.

*3.1.1 **Prediction**.* Participants first observed the scene with the robot and containers along with the current pouring parameters on a screen interface (Fig. 4) and visual condition assigned. We expected participants to consider several factors to make their decision on whether the robot will cause a spill during its pour action. These factors can include their intuition on the fluid dynamics of water along with the geometry of the source container, the horizontal and vertical distance between the containers, rotational velocity, and pour angle limit. No pour action is carried out in the first part of the task. The participant's prediction was recorded and compared against the ground truth which was determined for each configuration prior to the experiment by performing the pouring action without participants in the room. All Participants had access to a desktop monitor next to the table that displays the robot's current pouring parameters along with their assigned visualization tools. Participants complete part one of the task after reporting what portion of the ten configurations were spills or no spills along with their confidence rating.

*3.1.2 **Parameter Adjustment**.* In the second part of the task, participants had the opportunity to make adjustments to the robot's pouring behavior to achieve a pour without causing a spill; however, participants could only make 1 parameter change before executing the pouring action. To clarify, if the participant changed the rotational velocity that was initially *Slow* to *Fast* but later wanted to change the Pour angle limit instead, then the rotational velocity parameter returned to the original configuration of *Slow*. The pour action is then executed after the participant is satisfied with their change. The research coordinator then measured the amount of fluid transferred with a digital scale while the participant answered questions regarding their experience in their recent round.

For this study, five of the pouring configurations results in a confirmed spill unless one of the four parameters is changed to a specific value. For example, a container's rotational velocity is initially set to "fast" which purposely resulted in a spill. To avoid the spill, the participants could change the velocity setting to "slow". Changing any other parameter will still result in a spill but could

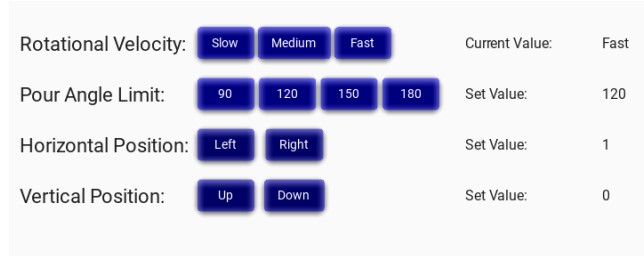

**Figure 4: Screenshot of the control panel used to adjust the parameters of the robot's pouring behavior .**

vary in the amount of fluid transfer. Once the participants were content with their changes, the research coordinator executes the robot's pour action and records the outcome. Participants then answered questions regarding that container. The source container then cycles between other common containers (See 3.3.4 for more details).

## 3.2 Pouring Parameters

- **Rotation Velocity** - refers to the percentage of the robot's maximum rotational velocity of the robot's wrist. Participants can control the speed by selecting the speed setting (slow, medium, fast)
- **Rotation Angle Limit** - refers to the limit on how far the wrist can rotate. Users can set the angle rotation by clicking one of the 4 options (90, 120, 150, 180)
- **Horizontal Position** - refers to the robot's end effector horizontal position. By clicking either the increase or decrease buttons, the robot moves along the horizontal axis approximately 1.5 cm respectively.
- **Vertical Position** - similar to the Horizontal Position parameter but along the vertical axis.

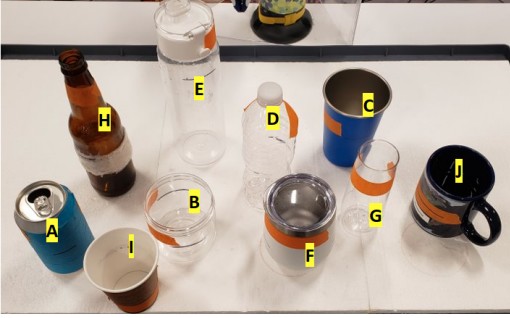

**Figure 5: The 10 source containers used in this study.**

## 3.3 Conditions

The following conditions were assigned to participants.

*3.3.1 **No Visuals (Controls)**.* In this condition, participants had no visualization device (AR or Rviz). Participants could only rely on what they can see of the robot and the current pouring parameters.

Participants could not visualize the future planned trajectory of the pouring behavior. The *No Visuals* condition evaluates **H1**.

*3.3.2* ***RViz (2D Screen)***. In this condition, participants were provided a standard monitor screen that runs RViz [28], a commonly used visualization tool in the robotics community. The virtual environment closely resembles the physical environment containing a digital twin of the table, robotic manipulator, and the two containers (Fig. 3 (E)). No fluid is simulated within the cup. Users can watch the simulated planned trajectory of the pouring behavior at any viewpoint by manipulating the camera view in RViz.

*3.3.3* ***Augmented Reality (AR)***. In this condition, participants used an AR device (Hololens2) to visualize the digital twin of the robotic arm and source container simulating the pouring behavior for the set pouring parameters. The visualizations in this condition mirror the robotic arm and source container visualized in the RViz condition. As users adjust these parameters, the visualizations update to show the new expected pouring behavior. No fluid is simulated within the cup and no additional virtual objects were rendered. Simulating fluid dynamics in our AR devices is not fully supported yet and is rather difficult to ensure the same fluid simulations from RViz matches the behavior in the AR device.

*3.3.4* ***Source Containers***. Ten source containers were chosen as common containers that people interact with in their daily lives that was distributed equally for rim size (i.e., wide rim, narrow rim) (Fig. 5). For this study, the research coordinator arbitrarily selected half of the containers to require one of the four pouring parameter changes. In the final selection, container (B) required changing the speed, container (E) required an angle limit change, container (F) required changing the Horizontal Position, and containers (H and J) required changing the Vertical position, with J seen as an obvious decision due to the proximity to the source container. The containers A, C, D, G, and I did not require any changes; note that a few containers result in a pour without spill even if the vertical parameter is adjusted. All containers had markings made by the researcher to indicate the where to place each container within the robotic gripper to ensure consistency with each pour.

## 3.4 Participants

We recruited a total of 24 participants (11 males, 13 females). Participants' age ranged from 19-41 with an average of 25 (SD = 5). The study took approximately 60 minutes for each participant to complete and consisted of 5 phases: 1) Introduction, 2) Calibration, 3) Demonstration, 4) Task, and 5) Post Survey. This study was approved by the university's IRB (STUDY00004011).

In the introduction, participants, through appointment with the lab coordinator, arrived at the study space and were given a consent form which they read and signed. Participants then reported on a five-point scale [0-4] pre-questionnaire their familiarity and usage with robots (M = 2.3, SD = 1.57) and augmented reality (M = 0.54, SD = 0.83). The lab coordinator then gave a brief explanation on the objective of study and the devices they were going to use. Participants who were selected to use the AR device were fitted with the Hololens2 and completed the onboard eye calibration process. In the demonstration phase, participants were shown both successful and unsuccessfully pour actions through a video and how each

parameter affects the robot's pouring behavior. For those with the visualization condition, participants could see the corresponding parameter changes in RViz and/or AR device. Participants then moved on to the main task phase, described in detail in Section 3.1. Finally, a post survey was issued that gauged their preferences in a Likert Scale of 10 items in a seven option response format and open-ended responses.

## 3.5 Measures & Analysis

We gathered a combination of objective and subjective measures to evaluate our hypotheses. In the *Prediction* portion of the study we collected:

> **Task success rate in spill prediction** - This measure is defined as the number of correct predictions divided by the total number of predictions.
> **Confidence Ratings** - This measure is a seven-option response format to "How confident are you in your answer?"

In the *Parameter Adjustment* portion of the study we collected:

> **Percent average of fluid transfer** - This measure is defined as the amount of fluid within the target container divided by the initial amount of fluid within the source container.
> **Parameter Selection Success Rate** - This measure is defined as the number of correct parameter selections over the total number of parameter selections. It is important to note again that containers A, C, D, G, and I do not require any parameter changes. There is a situation that we have noted where if a participant were to change the *Vertical* parameter for these containers, then the pour outcome may not result in a spill. In this scenario, we would still mark their *Parameter selection* as incorrect.
> **Confidence ratings** - This measure is a seven-option response format to the following: "It was difficult to pick which parameter to adjust to prevent a spill.".
> **Open-ended Responses** - This measure is an open-ended response to the following: "If a spill did occur, what would you have done differently?" and "How did you select which parameter to change?".

The post-survey included a Likert Scale with 10 Likert items with seven option response format (Cronbach's Alpha, $\alpha = 0.95$) that gauged preference between the participant's two visual conditions. Items included the following where Condition 1 and 2 are placeholders for their assigned conditions:

- (Condition 2) is easier to learn and use than (Condition 1).
- I find (Condition 2)'s interface more user-friendly compared to (Condition 1).
- (Condition 2) helped me complete my task more efficiently than (Condition 1).

Three open-ended responses aimed to gather additional information to describe their experience with the visual condition and source containers:

- "What were some signs that led you to believe that the robot was going to cause a spill or not?"
- "What other types of information about the robot or containers not provided to you in this study that could've helped you determine the outcome of the pour?"

- "Overall, what type of containers did you believe were the easiest to predict? Hardest to predict?"

We analyzed data using a one-way Analysis of Variance (ANOVA) with experimental condition of Visualization as the fixed effect. Standard Post-hoc tests, including Tukey's Correction test, compared the effectiveness across each visual condition. P-values were adjusted using the Bonferroni Correction.

## 3.6 Hardware & Software

We used a 6-DoF *Kinova Gen3 Lite* robot with a 2-finger gripper controlled with *Robot Operating System* (ROS) [40] and *MoveIt* [11] running on Ubuntu 18.04. Python and C++ scripts filtered the robotic data before delivery to the AR device and RViz [28] for visualization. Augmented visuals were developed with *Unity 2020.3.44f1*[1] and deployed onto a *Hololens2* which rendered the data over the physical environment. *Vuforia*[2] tracked the robot's position and orientation through a cylindrical target-image fixed to the robot (see Figure 1). A digital-twin of the robot provided by *Kinova*[3] is placed with the target-image to match the real robot setup with C# scripts that control data exchange between the robot and AR device that occurred over a shared Wi-fi network using *ROS-TCP-Connector*[4]. Scaled digital twins of the source containers were created in modeling software and imported to both Unity and RViz. The experiment took place in an isolated lab space that included a robotic manipulator and a target container fixed to a standard table. The amount of water filled in each source container was approximately half the container. A desktop computer connected to the robot was stationed next to the table with a tablet displaying the UI panel to control the robot pouring behavior (see Figure 4) along with a separate window to record participants' questionnaire responses.

## 4 RESULTS

We analyzed the accuracy of *Spill Prediction*, i.e., the extent to which the participant was able to to correctly determine whether or not the robot will cause a spill given the set pouring parameters. *No Visuals* had an average score of 51.3% (SD = 26.3), *RViz* had an average score of 52.5% (SD = 17.7), and AR had an average score of 53.8% (SD = 22.8). We found no significant main effect $F(2, 45) = 0.05, p = 0.95$ rejecting **H1**.

We evaluated participant's confidence in the *Prediction* portion of the study. *No Visuals* had an average rating of 5.14 (SD = 0.64), *RViz* had an average rating of 5.09 (SD = 0.72), and AR had an average rating of 5.01 (SD = 0.67). We found no significant findings as well $F(2, 45) = 0.14, p = 0.87$. All source containers averaged along "Somewhat Agree" in response to the difficulty rating, "It was difficult to pick which parameter to adjust to prevent a spill."

As for the success rate of *Parameter Selection* in which the participant makes the correct parameter selection, *No Visuals* had an average score of 26.3% (SD = 15.9), *RViz* had an average score of 32.5% (SD = 16.1), and AR had an average score of 41.3% (SD = 23.6). We found a marginal main effect $F(2, 45) = 2.55, p = 0.09$ not fully

supporting **H2a**. Confidence ratings for parameter changes showed no main effect $F(2, 45) = 0.93, p = 0.40$ not supporting **H2c**.

Finally, we analyzed the parameter section by container type Narrow (N) vs. Wide(W). *No Visuals* had an average score of 27.5% (SD = 29.8) and 25.0% (SD = 29.3) , *RViz* had an average score of 32.5% (SD = 25.9) and 32.5% (SD = 19.0), and AR had an average score of 50.0 (SD = 31.9) and 32.5 (SD = 11.2) for N and W respectively. We found no significant main effect $F(2, 24) = 0.34, p = 0.71$ rejecting **H3**. Figure 7 shows the average score on *Parameter Selection* by visual condition and container type (Narrow vs. Wide).

Figure 6 shows the average percent of fluid that was transferred to the target container. *No Visuals* had an average score of 78.6% (SD = 7.27), *RViz* had an average score of 79.5% (SD = 8.97), and AR had an average score of 86.5% (SD = 11.3). Overall performance revealed a significant main effect by visual condition, $F(2, 45) = 3.46, p < 0.04$ with an Effect size = 0.09 suggesting that the visualization condition accounts for a moderate proportion of the variance in fluid transfer. Comparing conditions of *No Visuals* and *RViz* to AR with Dunnetts's multiple comparison test, we found a decrease in performance with *No Visuals*(CT) ($p = 0.04$) supporting **H2b**.

If we look into individual containers that required specific parameter changes, we find significant effects for containers D, E, and F. Container D, which required no changes with the water bottle showed *No Visuals* had an average transfer of 93.5% (SD = 10.9), *RViz* had an average transfer of 66.3% (SD = 35.6), and AR had an average transfer of 96.7% (SD = 1.66). AR enabled users to see that the pouring parameters were adequate for a successful pour $F(2, 21) = 4.82, p = 0.02$. Post-hoc comparisons revealed that AR significantly improved perceptions of the pouring behavior over *RViz* ($p = 0.02$). Container E, which required an angle limit change with the sports bottle for a successful pour showed *No Visuals* had an average transfer of 31.4% (SD = 28.0), *RViz* had an average transfer of 72.2% (SD = 37.0), and AR had an average transfer of 73.4% (SD = 35.3) and a main effect $F(2, 21) = 4.03, p = 0.03$. Post-hoc comparisons using Tukey's Honestly Significant Difference (HSD) revealed that both AR and RViz marginally ($0.10 > p > 0.05$) improved perceptions for angular change in pouring outcomes ($p = 0.05, p = 0.06$) respectively. Container F, which required a horizontal change with the tumbler for a successful pour showed *No Visuals* had an average transfer of 43.8% (SD = 23.9), *RViz* had an average score of 35.9% (SD = 10.1), and AR had an average score of 67.3% (SD = 24.9) and a main effect, $F(2, 21) = 4.95, p = 0.02$. Post-hoc comparisons using Tukey's HSD revealed that users had a greater perception for horizontal positioning compared to RViz ($p = 0.01$) and a marginal main effect compared to *No Visuals* ($p = 0.83$).

We also ran a Sample Size calculation with a target Power = 0.8, a significance level $\alpha = 0.05$, and the effect size $\delta = 0.13$ to determine the number of participants we needed to run was approximately 68, which is well above the total number we recruited.

## 5 DISCUSSION

There could be a number of reasons why we did not find a significant effect for **H1**. Participants on average were not highly experienced with AR or robotics as reported with the pre-questionnaire mentioned in Section 3.4. Because this study may be a participant's first experience with augmented reality and RViz, they may not

---
[1]https://unity.com/
[2]https://developer.vuforia.com/
[3]https://github.com/Kinovarobotics/ros_kortex
[4]https://github.com/Unity-Technologies/ROS-TCP-Connector

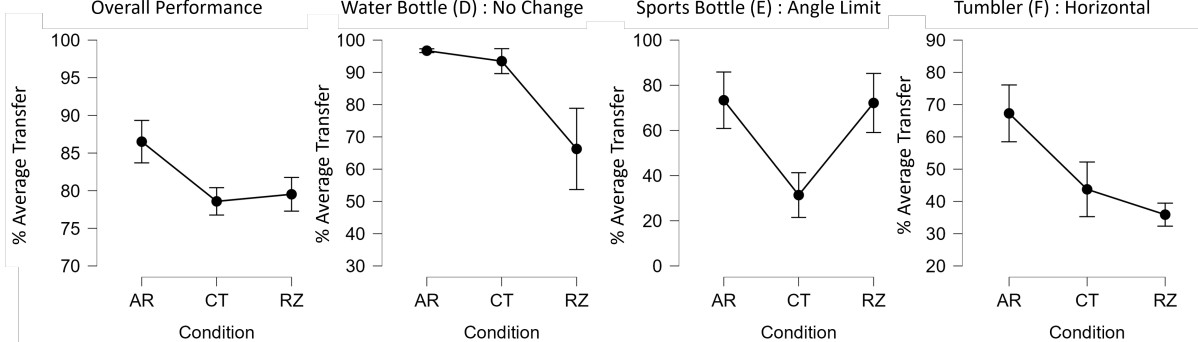

Figure 6: Average percentage fluid transfers by visual condition. T-bars indicate standard error.

be confident in using either device right away. The reported confidence in spill prediction averaged along "Somewhat Confident", so it is worth testing to compare inexperienced with experienced users to see if experienced users make better spill predictions. Additionally, the pouring configurations and various containers used in our study could be too challenging for users to work with as observed in the Parameter Selection Difficulty rating. Participants have mentioned both wide and narrow containers as difficult containers to predict their outcomes. For example, one user reported the following when asked about the difficulty of the task associated with specific containers:

> **[P:6 (RZ/AR)]** *"Easier: mugs with wider openings Harder: taller bottles with lids"*
> **[P:7 (RZ/AR)]** *"Easiest: Tall, narrow containers (openings and water trajectory were easiest to see when animated). Hardest: Short, wide containers."*

These opposing views in participants' belief to which containers make the task easier or more difficult could explain why we did not find a significant difference for **H2c**.

Although participants were not accurate in selecting the correct pouring parameters to achieve a successful pour for (**H2a**), visuals may have helped participants position the robot to minimize the distance between the two containers ultimately allowing the bulk

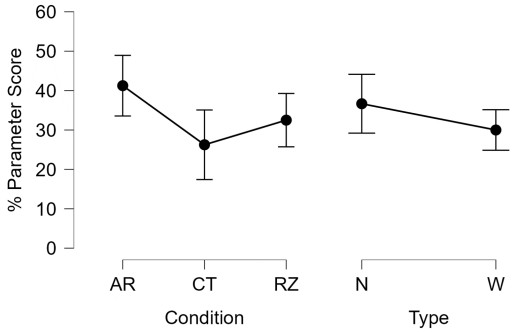

Figure 7: Average score in *Parameter Selection* by condition (Left) and by container type (right) where N and W correspond to Narrow and Wide respectively.

of the water to transfer to the target container **H2b**. We noted that most adjustments users made consisted of changing the horizontal or vertical position of the end effector.

> **[P:17 (AR/CT)]** *"For the non-ar case I based my decisions mostly off of intuition. The AR did a good job of showing how the two containers would align, using this I adjusted the robot until the two containers aligned in a way that intuition dictated would not spill."*
> **[P:24 (CT/RZ)]** *"The signs that I looked for were, 1) is the container positionally too close to the cup, or 2) will the pour-angle work for to pour all the liquid."*

Some participants mentioned relying on personal experience to make their decisions.

> **[P:12 (RZ/CT)]** *"The ones [containers] I use everyday like the water bottle that you get at the supermarket and the paper and solo like cup and the mug and soda can and bottle. I don't really use a sport water bottle or a tumbler so I am not used to how they pour."*
> **[P:22 (CT/RZ)]** *"RViz helped me mentally visualize the action and personal experience regarding the container attributes."*

A possible reason for the poor performance in the RViz condition is relative scaling of the objects within the virtual environment. Few participants suggested that the visuals in both AR and RViz(RZ) may be slightly misaligned:

> **[P:4 (RZ/CT)]** *"No Visuals [Control] was easier for me to understand because I felt that RViz offered somewhat conflicting views between the simulation to the real robot...".*
> **[P:20 (CT/RZ)]** *"No visuals [Control] allowed me to rely on my intuitions when predicting a spill. In contrast, RViz seemed a bit confusing and caused me to contradict myself."*

Such misalignment could explain why differences were found for container D that did not require any changes to result in a pour without spill. However, we noticed that participants seemed to rely solely on the visuals when assigned either the AR and RViz condition. We observed a few participants of the *No Visuals* condition using their hands as a makeshift ruler to determine if the lip of the source container will reach the target container by rotating their

wrists with a finger fixed at the center of the rotation point on the gripper. Although we did not expect this strategy, we allowed participants to continue and use any other strategies that they could use to make a better decision as long as they did not physically touch the robot or container.

> **[P:21 (CT/AR)]** *"The trajectory in AR was really helpful, which I could only imaging or measure by hands with No Visual."*

This strategy of using fixed lengths to gauge distance could explain the greater performance by the No Visual condition compared to RViz. We also noticed that participants seemed to rely mostly on the visuals when assigned to the AR and RViz condition. Although they have the opportunity to view the scene, we did not record any eye tracking data to gain insights to determine where and for how long their focus was when making their decisions.

Participants highlight advantages of AR over the 2D display(RZ) and *No Visuals* conditions which contribute to the overall performance in fluid transfer:

> **[P:17 (AR/CT)]** *"AR was very useful to show where the container would end up at the end of its cycle."*
> **[P:7 (RZ/AR)]** *"It was useful seeing the animation directly overlaid with the object in real life, which helped me make more accurate predictions."*
> **[P:10 (CT/AR)]** *"Actually visualizing it was super useful, and I found myself doing that mentally for no visuals"*

Although the *Hololens2* has shown its usefulness in mental alignment as an AR device, participants have pointed out disadvantages which could have contributed to low subjective ratings.

> **[P:11 (AR/RZ)]** *"Rviz does not need calibration so it is much easier and also it is more user friendly to people wearing glasses"*
> **[P:10 (CT/AR)]** *"The headset felt clunky, and the depth perception made me feel a little off. No visuals felt more free"*
> **[P:2 (AR/CT)]** *"The headset is kinda uncomfortable"*

Participants assigned to both *RViz*(RZ) and*No Visuals*(CT) conditions stated the usefulness of having visuals demonstrating the pouring behavior.

> **[P:1 (RZ/CT)]** *"I like how I can see what the animation will look like before running it."*
> **[P:3 (RZ/CT)]** *"In No Visuals, I could imagine the motion of the container but it was harder to visualize how that motion would occur in relation to the cup. RViz made this much easier."*
> **[P:22 (CT/RZ)]** *"I like that RViz loops through the speed of which the robot is rotating"*

However, a few participants stated why No Visuals was more beneficial for them other than the alignment issue mentioned earlier.

> **[P:21 (CT/AR)]** *"No setup, no calibration, easier"*
> **[P:5 (CT/RZ)]** *"no visuals is straightforward and doesn't require looking in two places"*

## 5.1 Limitations & Future Work

Although we demonstrated a potential application of AR and 2D screens (RViz) for robotic pouring, there were some limitations. The target container was fixed relative to the robot arm along with its digital twin within RViz for simplicity. Ideally the robot should perceive both the source and target cup to account for any misalignment so that RViz displays a more accurate representation of the pouring scene. We can only say that the virtual scene created in RViz was matched to the real world as close as possible. This drawback highlights the advantage of AR by only needing to render the source container instead of the entire virtual environment. Another limitation for our study is implementing a partial within study design where participants were assigned to two out of three visual conditions. A full within participant study where each participant would see all visual conditions and source containers using balanced Latin squares to account for ordering-effects would have been infeasible due to the number of required participants; therefore, we decided to compensate with the partial within study design. We focused on controlling the visual conditions rather than the type of source containers and sacrificed order-effects among the source containers.

While participants in this study did not engage in direct interaction with the robot, we showcase how AR holds promise in enhancing users' predictive abilities regarding pouring outcomes. Future research could also delve into user engagement with the robot, directing fluid manipulation under defined pouring conditions while observing parameter alterations through AR. Another direction includes a similar study with simulating fluids as new visualizations techniques to convey the robot's pouring intent [38]. Sixty percent of participants claimed to have wanted a form of fluid simulator and water level indicator as additional useful information.

Our results showed that participants had greater performance when adjusting the angular limit and horizontal positioning; however, it is not clear if the geometry of the container played a role. It would be worth testing pouring behaviors with other container shapes such as rectangular containers [29].

## 6 CONCLUSIONS

In this work, we explored how visualizing a robot's planned actions could improve a person's prediction on the outcome of that action in the context of pouring behaviors. We used pouring as a task that is intuitive for people but challenging for robots. In a two-part task, participants first predict the outcome of a pouring behavior from various source containers and then adjust the robot's pouring parameters to reduce spillage. We found that visualizing the robot's future planned pouring trajectory in augmented reality (AR) significantly improved a participant's ability to reduce spills. Pour angle limit and horizontal positioning were two key parameters that participants were able to discern as faults and correct to avoid a spill event. However, spill prediction remained a challenge for all participants. Visualizing the planned pouring trajectory with fluid simulators in future work may improve user prediction of pouring outcomes for assistive robotics. We hope this study will inspire and inform future research about how different modes of visualization can improve the abilities of robots to interact with liquids in assistive robotics settings.

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
