# OpenReview forum: "Helping People Predict the Outcome of Robotic Pouring Behaviors with Augmented Reality"
_humanrobotinteraction.org/HRI/2024/Workshop/VAM-HRI — VAM-HRI 2024 Oral_

### Official Review · Reviewer_aTDz · 2024-02-02
**Review1**

**Rating:** 8
**Confidence:** 5

**Review:**

The paper is about user studies of the use of cobots for pouring fluids from one container (cups/bottles) to another to prevent spills. The paper compares different UI systems such as 2D displays or AR headsets.
Papers is a strong and interesting user study, that can be a step towards assistive in-home robots. I enjoy reading it and have some comments that can enrich the paper.

1. While now it is a bit too late, it is always a good practice to register your hypothesis beforehand on e.g. OSF platform.
2. It would be nice to get a bit more insight about different pouring configurations and how they were set up.
3. I think a big downside of this project is the lack of direct interaction with robots. Users observed the robots pouring the liquid, they can readjust some settings, etc, but they do not control the robots per se. That makes this paper a bit weaker. I think the complete study would be that after observing and being able to adjust some parameters, users could use AR glasses, arrows on the keyboard, or a joystick to control the robot while pouring the liquid. My intuition is that this would make the paper complete.
3b. If we would only want to visualize the movement, it would be nice to get a bit more complex behavior, not only 1 joint movement.
4.  Good related work but it would be good to look at and acknowledge other studies that deal with AR/VR and robots/cobots and/or compare different UIs (VR vs. screen) with cobots (last year RO-MAN and HRI). It could be quite useful to review it as well:
- Happily Error After: Framework Development and User Study for Correcting Robot Perception Errors in Virtual Reality
https://ieeexplore.ieee.org/document/10309446- What You See Is (not) What You Get: A VR Framework for Correcting Robot Errors
https://dl.acm.org/doi/abs/10.1145/3568294.3580081
- Reimagining RViz: Multidimensional Augmented Reality Robot Signal Design
https://ieeexplore.ieee.org/abstract/document/9900692
- PhysicalTwin: Mixed Reality Interaction Environment for AI-Supported Assistive Robots
https://openreview.net/forum?id=dz0EoKeBFW
- Empowering Cobots with Energy Models: Real Augmented Digital Twin Cobot with Accurate Energy Consumption Model
https://ieeexplore.ieee.org/document/10309614

5. It's always nice to make your solution open source, if possible :)
6. Good results analysis, I would vote for a bit bigger plots, if the space is not restricted

---

### Official Review · Reviewer_DEmc · 2024-02-03
**Review 2**

**Rating:** 8
**Confidence:** 5

**Review:**

This paper introduces an augmented reality system to assist users in setting parameters for robotic fluid-pouring tasks, by visualizing the pouring motion holographically in environmental context. The authors conduct a human subjects study, comparing performance on human-assisted pouring tasks between three conditions: no added visualization, 2D display digital twin-based visualization, and augmented reality-based visualization. The results of the user study were mixed, showing some limited prediction and performance improvements when moving to AR-based visualization.

Strengths:
- The paper is well motivated and clearly situated in the state of the art, aiming at a simple, specific problem where autonomous robots struggle, and providing an interface to allow for targeted human feedback to address that problem.
- The user study's task and experimental design are both very thorough. Measuring over 10 distinct source containers is a helpful addition, due to the vastly different pouring dynamics of each container which is intuitive to human users but difficult to model in simulation. Likewise, the statistical analysis is well-done and thorough.
- The paper as a whole is very well written, with high clarity and an easy to follow narrative.

Weaknesses:
- The justification for the specific experiment design is unclear at the beginning of Section 3 (the 3 x 10 partial within-within participant study). Why were participants assigned two out of the three conditions, rather than the simpler full within or full between? Brief discussion of this is warranted. Similarly, I'm confused as to whether participants see the control (no visualization) condition, in addition to one of the two visualization-based conditions, or if the conditions are fully randomly selected.
- Locking participants to a single parameter change could use some explanation - I understand, for simplicity's sake, designing the tasks so that one parameter change could lead to a successful pour, but why will the system revert previous parameter changes if the participant attempts to make more than one?
- More description is needed on how the parameters are grounded in the no visualization condition. How do participants know what the horizontal/vertical position numbers represent spatially, or what slow vs. fast rotational velocity looks like? Do they receive a pre-experiment briefing on how to interpret this data? Without grounding, I think it would be unfairly difficult for a novice user to use such a system.

Review Summary:
This paper would be a fantastic addition to the program of this year's VAM-HRI, and I believe it will generate some good discussion on how such an AR system could be altered to increase its effectiveness for real-life pouring tasks. As such, I recommend acceptance to VAM-HRI 2024.

---

### Decision · Program_Chairs · 2024-02-10

Accept (Oral)